# Analysis of Housing Risk Factors for the Welfare of Lean and Heavy Pigs in a Sample of European Fattening Farms

**DOI:** 10.3390/ani11113221

**Published:** 2021-11-11

**Authors:** Paolo Ferrari, Alessandro Ulrici, Matteo Barbari

**Affiliations:** 1Department of Agriculture, Food, Environment and Forestry (DAGRI), University of Florence, P.le delle Cascine, 18, 50144 Firenze, Italy; matteo.barbari@unifi.it; 2Department of Life Sciences, University of Modena and Reggio Emilia, Via Amendola, 2, 42122 Reggio Emilia, Italy; alessandro.ulrici@unimore.it

**Keywords:** housing system, pig welfare, fattening pig, body lesion scores, bedding material, enriched environment, roughage, tail docking

## Abstract

**Simple Summary:**

Animal welfare is a major challenge that most European pig producers have been facing in recent decades to comply with EU legislation and to meet the increasing societal and market demand for pork produced in a sustainable way. Pig welfare is ruled in terms of minimum requirements for housing and management, but stakeholders have considered that both farm-level and animal-based indicators are fundamental to monitor animal welfare. Some of the welfare issues still affecting fattening pigs are the lack of space, bedding and manipulable material, and the continued practice of routine tail docking of pigs. Tail docking is applied routinely across most European countries to reduce the occurrence of severe tail biting lesions, despite its ban in the EU. An observational study on 51 pig farms in seven EU countries, aimed at investigating housing risk factors for the welfare of finishing pigs, showed that body weight and presence of bedded solid floored resting area (BED) identify three clusters of farms. The outcomes of this study confirmed that BED and larger availability of space per pig, above the minimum requirement of EU legislation, can limit the occurrence of lesions in pigs with undocked tails.

**Abstract:**

Pig welfare is affected by housing conditions, the minimum requirements of which are set up by EU legislation. Animal and non-animal-based measures are useful indicators to investigate housing risk factors for pig welfare. An observational study on 51 pig farms in seven EU countries, aimed at investigating housing risk factors for the welfare of finishing pigs, showed body weight and presence of bedded solid floored resting area (BED) identifying three clusters of farms. Farms with BED were featured by no or limited tail docking, larger availability of manipulable materials and lower number of pigs per farm and per annual work unit. In these farms, less skin and ear lesions were found, compared with lean pigs of farms without BED, which were characterized by lower pig space allowance, mortality rate and medication cost. In farms without BED, heavy pigs were featured by more space per pig, more pigs per drinker and higher mortality rate and medication cost per pig, compared to lean pigs. No statistical difference in tail lesions was found between the three farm clusters, although tail docking was performed in all farms without BED and not performed on most farms with BED.

## 1. Introduction

Animal welfare is a major challenge that most European pig producers have been facing in recent decades to comply with EU legislation and to meet the increasing societal and market demand for pork produced in a sustainable way. Although pig welfare has been governed by EU legislation since 1991 [1], some major welfare issues still remain, such as the lack of space allowance, enrichment materials and bedding, and the practice of tail docking if carried out routinely [2]. Housing conditions are deemed by stakeholders as particularly important to safeguard animal welfare, as well as the use of animal-based and farm-level indicators to monitor the progress of animal welfare [3].

Animal-based measures were developed to directly assess the effective welfare state of the pig by measuring, for example, its behaviour, fearfulness, health, or physical condition [4]. Nevertheless, the European legislation for the protection of pigs is based on housing and management risks which can be assessed by using resource-based measures (i.e., non-animal-based measures) rather than animal-based measures [5]. Resource-based measures are indirect measures of animal welfare because measuring the ability of the farming system (housing and management) to provide pigs with conditions to which the pigs can adapt without endangering their welfare; therefore, monitoring resource-based measures can be useful to identify risk factors that lead or may lead to actual welfare problems in pigs, which can be measured by animal-based measures. Therefore, monitoring both animal-based and non-animal-based measures is a promising approach to advice pig farmers to control and improve the welfare conditions of pigs. “Age of the animals, type of floor, feeding system, stocking density and environmental temperature can be useful to predict the appearance of a given welfare measure of ‘good housing’ on a farm” [6,7].

European legislation sets to 1 m^2^/pig the minimum floor space allowance for pigs over 110 kg of body weight without any further indication for heavier pigs, such as the Italian heavy pigs slaughtered at the minimum age of 9 months and at the live weight of 160 kg ± 10%, according to the Parma ham Protected Designation of Origin (PDO) scheme [8]. The allometric equation A = k × BW^0.67^ was used with k = 0.03 to calculate the minimum legal space allowance for growing and fattening pigs [9] although EFSA recommended k = 0.036 for pigs up to 110 kg of live weight and k = 0.047 above 110 kg, to allow all pigs to rest simultaneously in lateral lying posture [10]. One study showed that less stocking densities and reduced pen size can lead to more pigs laying at the same time, less pig lesions, less pen dirtiness and higher average daily growth [11].

The influence of group size on pig welfare is controversial; no significant effect was proved on fattening pigs according to some authors [12,13,14,15], whereas an increase in group size would result into unfavourable effects on welfare and performance, according to other studies [11].

Pig dunging behaviour is affected by space allowance because different functional areas are used by pigs for resting and for dunging, unless the pigs are heat stressed or sick or the stocking density is too high [16]. Pig and litter cleanliness in straw bedded pens was also found as negatively affected by hot climate [17]. One study showed that pen soiling increases with increasing age in pigs kept on solid floor, as floor soiling and wallowing behaviour was more prevalent in the late growth period [18]. More pig soiling was observed in pigs that were liquid fed, compared to pigs fed with wet and dry feed [7]. Italian heavy pigs are traditionally liquid fed during the entire growing-fattening phase, which is approximately 6 month long [19].

Free access to water of good quality is mandatory in fattening pigs [20] and needed even if liquid or wet feed is provided [16]. To this end, a maximum number of fattening pigs per functioning drinker is recommended, depending on the type of drinker [21]: 12 pigs per nipple or 15 pigs per water bowl.

Slatted floors systems are widely used for pig housing throughout the EU [16]. Fully slatted or partially slatted floors are generally used to house heavy pigs in Italy [19]. They promote pig cleanliness and hygiene by allowing the quick and effective removal of faeces and urine from the pen, although fully slatted floors were found to limit the use of straw as bedding or manipulable material to allow pigs to perform explorative behaviour [16].

Manipulable materials are needed to enrich the pen environment for pigs intensively kept in order to meet their exploration behavioural needs and avoid tail biting and skin lesions in growing and fattening pigs kept at high stocking densities [22]. One study found decreased exploration of enrichment material with increasing live weight [15], so special attention should be given to providing effective enrichment as the pigs’ live weight increases.

Enrichment materials are categorized as [23]: (a) optimal materials which can be used alone because they are “edible, chewable, investigable, manipulable, of sustainable interest, accessible for oral manipulation, given in sufficient quantity, clean and hygienic”; (b) suboptimal materials, possessing most of the previous characteristics but not all of them so that their use should be combined with other materials; (c) “materials of marginal interest providing distraction for pigs which should not be considered as fulfilling their essential needs”.

Straw is considered as one of the best enrichment materials [16]; it has been demonstrated that its distribution in racks in fully slatted housing systems is possible and does not compromise the effectiveness of the manure removal system [24].

Pig welfare is also affected by stockperson’s action [16], so the ratio of number of pigs to number of stockpersons was acknowledged as a predictor variable for severe tail lesions in heavy pigs [25].

Most pigs in Europe are tail-docked despite the fact that the practice of routine tail-docking was banned in 1994 [1]. Tail docking aims at reducing the frequency of tail biting and the related tail lesions, but it is painful for pigs and can lead to neuroma formation [26].

However, the occurrence of tail biting depends on a wide range of factors such as the lack of environmental enrichment, stocking density, presence of slatted floors, microclimate discomfort, high levels of dust and noxious gases (i.e., ammonia), competition for resources, social instability and genetic, dietary and health factors [14,26]. Additional risk factors to predict farms having severe tail lesions were identified in: pig age, live weight at slaughter, space allowance for 100 kg of live weight, number and type of drinkers, pen size and number of pigs in the farm [22,25].

Tail biting can occur in all production systems, including free-range and organic [27,28,29]. Particular attention should be paid by farmers keeping pigs with intact tails through frequent observation and timely intervention in case of tail-biting outbreaks, which can spread rapidly and become difficult to stop [30].

The prevalence of physical conditions in pigs varies between herds [27,31]. Tail, skin, and ear lesions are used widely as animal-based measures to directly assess animal welfare of growing and fattening pigs [4,5].

The farm average pig’s mortality rate is a common measure of health and welfare for pig herds. Mortality is defined as “the uncontrolled death of animals (as distinct from culling/euthanasia). Any animal which is found dead on the floor in the house, or out on the field is considered a mortality” [4]. Pigs may be culled (i.e., emergency killing) if they are injured or sick to avoid exposing them to severe pain or suffering, or if no other practical way is available to relieve the pain [32]. One study shows that emergency killing is more frequently implemented on piglets rather than on older pigs, such as growing and fattening pigs [33].

## 2. Materials and Methods

An observational study was carried out across seven EU countries by using the Condensed protocol from the Era-Net SusAn project “Sustainable pig production systems” (SusPigSys) [34]. Animal welfare data, together with a number of economic/production data considered as relevant for animal welfare assessment, were included in this study for growing-fattening units involved in the SusPigSys project. For this purpose, 31 non-animal and animal-based measures were considered as animal welfare indicators for both heavy and lean pig farms in the growing and finishing phase, across fattening units of 51 pig farms in seven EU countries (i.e., IT, DE, AT, NL, PL, FI, UK). Type and description of variables are given in Table 1 and Table 2. Farm units with uncomplete data were excluded from this study, as well as cases of observed pig groups with uncomplete or inconsistent animal welfare data for one or more variables. Complete data from 51 fattening pig units were processed statistically, including: (1) economic/production data of visited farms, considered as potentially relevant for pig welfare (i.e., affecting or affected by pig welfare); (2) animal welfare measures of up to 15 pig groups observed during farm visit, except for three Polish farms in which 16, 17 and 18 pig groups have been observed for more representativity.

Pig group sampling and pig number sizing in each group for collection of animal welfare data were based on the “Real Welfare” scheme strategy [35].

Pig groups were chosen randomly and in proportion of their stage of production: one third of the pig groups at the early fattening period but grouped at least two weeks before farm visit, one third in the middle of the fattening period and one third at the end of the fattening period. For pig groups with 100 or less pigs per group, up to 15 pig groups were assessed by observing up to 50 pigs per group. For groups larger than 100 pigs, at least 50% of pigs per group were observed up to a total of 750 pigs for all groups. Animal-based information was collected by observing the animals in their own environment from a distance of 50 cm. A total of 709 pig groups from the 51 pig farms were assessed.

Prior to the start of farm visits, training material including definitions was created and assessors were trained at a joint training occasion in order to achieve a consistent scoring, which was tested as inter-assessor agreement (IOR) on-farm for all measures that required scoring, using joint assessments and photo material. IOR was calculated as exact agreement between two observers and expressed as weighted Kappa, PABAK and percentage agreement.

Farmers were recruited on a voluntary basis; before and at the beginning of each farm visit, the farmer (i.e., person(s) to be interviewed) was informed in speech and writing about the project, including information about anonymity, why the research was being conducted, how his or her data were being used and if there were any risks associated, and were asked to return a signed informed consent before the start of data collection. Farmers were asked, before pig observation, about the number of pig houses, pig groups per pig house, pigs per group and related ages, and if no, or some, or all pigs were tail docked.

Twenty-three variables were taken into account for the observed pig groups (Table 1): 14 non animal based measures were considered as relevant for pig housing conditions, productivity and management (i.e., four continuous, seven ordinal and three dichotomous variables) together with nine animal-based measures (i.e., four continuous and five ordinal variables), as relevant for the presence or prevalence of pig lesions and for the pig behaviour towards manipulable materials.

Animal welfare measures on 709 pig groups were aggregated as mean values per farm for continuous variables or median values per farm for most ordinal variables, except for variables EP, T, E and B; for each farm the mean value or the median value of pig group observations were considered for each variable. Enrichment presence (EP), tail lesions (T), ear lesions (E) and body lesions (B), as described in Table 1, were transformed from ordinal variables (i.e., score 0, 1, 2) into continuous variables, as farm percentage of observed pens with optimal or suboptimal enrichment (EC) or with at least one pig with a mild or severe tail (T), ear (E), and body (B) lesion, respectively (Table 2).

The dataset of 51 cases of fattening units was obtained, including eight additional continuous variables (Table 2): six of them related to farm management and productivity, one related to mortality rate (i.e., not including culled, sick or unproductive pigs) in the calendar year before farm observation (i.e., 2018) and one related to the farm percentage of pig groups with at least 1 nipple drinker per 12 pigs or 1 water bowl per 15 pigs.

Management and productivity data were collected in an interview with the farm manager or the person responsible for pig care using the SusPigSys protocol [34]. Data on housing conditions in the pig houses were directly recorded.

As a pig stockperson may be full or part-time employed, the average number of pigs present on a farm (i.e., AVP) was related to the Annual Work Unit (AWU), as defined by Eurostat [36] for a stockperson occupied in pig farming on a full-time basis; this variable is the average number of pigs present on a farm in 2018 per AWU (PWU).

Statistical analysis of all data was performed with SPSS Statistics 27, except for Principal Component Analysis (PCA), which was performed using the PLS Toolbox software (v. 8.8.1). The dataset of 51 European pig farms was explored by means of PCA, in order to obtain an overview of the overall data structure, both in terms of correlations between the considered variables and of samples (farms) clustering. The loading plots were used to investigate the relationships between variables. A PCA model was calculated on the whole dataset of pig farms using autoscaling as the variable pre-processing method. Autoscaling consists of transforming each variable by subtracting its average value and then dividing it by its standard deviation. This transformation allows the data to be translated at the origin of the reference system, since each variable will have an average value equal to zero, and also makes the variability of each variable equally important in the construction of the PCA model, since each variable will have standard deviation equal to one [37]. Variables that are close to each other in the loading plot have similar properties and variables that are far apart and are different from each other [38]. The score plots were used both to highlight similarities and differences between the pig farms, and for direct interpretation of the farm cases in relation to variables in the loading plots.

Non-parametric analysis (Kruskal–Wallis test) was applied to the single variables, for not normally distributed data, to further explore the differences between clusters identified by the PCA. Furthermore, pairwise comparisons were performed using the Mann–Whitney U-test, when a significant effect of the farm group was revealed.

## 3. Results

The PCA resulted in four Principal Components (PCs), selected on the basis of the scree plot, explaining 58.43% of total data variance. The variance not captured by the model can be ascribed to statistical noise due to a number of possible factors such as the biological nature of most data and the limited sensitivity of some variables.

The loading plot of the first two principal components (Figure 1a) shows that PC1, accounting for the largest data variability, is mainly influenced by the variables lying at the left and right extreme parts of the plot.

In particular, on the right side, the following variables are positively correlated with each other and with PC1: bedding presence (BP), roughage presence (RP), pig access to enrichment (PAE), enrichment material behaviour (EMB) and percentage of pens with optimal or suboptimal enrichment (EC). These variables are negatively correlated with those on the left side of the plot: tail docked pigs (TD), tail stump pigs (STT) and slatted floors (SF).

The score plot of the first two PCs clearly shows three groups of farms located on the right side, on the left topside and on the left downside, respectively (Figure 1b). The group on the right is composed by the only six organic farms in the dataset and by four non organic farms, characterized by some welfare standards similar to the organic ones, according to Regulation (EC) 889/2008 (i.e., bedding in the resting areas, no or limited slatted floor, no or limited tail docking and high space allowance), and above the minimum requirements of Directive 120/2008/EC; this latter group includes two Polish, one German and one Finnish farm (Figure 2).

The group on the left topside consists of 10 out of the 12 Italian heavy pig farms (Figure 2), whose positions in the score plot, compared to the corresponding loading plot of Figure 1, reveals that they are characterized by higher live weight at slaughter (LWS), feed conversion ratio (FCR), mortality rate (M), number of pigs per drinker (PD) and the presence of liquid feeding system (LFS). The other two Italian heavy pig farms are organic and represented correctly on the right side of the score plot.

Based on the results of the PCA model, which highlighted the presence of three separate farm clusters (Figure 2), the contribution of each single variable to this grouping was explored by checking for statistically significant differences between clusters:Housing system with bedded solid floored resting area (BED);Housing system with no bedded solid floored resting area for lean pigs (NBL);Housing system with no bedded solid floored resting area for heavy pigs (NBH).

Pen size and number of pigs in the observed pens were not founded statistically different in the three farm groups (Table 3), whereas higher farm average pig live weight (AW) was observed (*p* = 2 × 10^−4^) in NBH farms than in NBL and BED farms, as expected. Lower space allowance per pig (SP) was found (*p* = 6 × 10^−6^) in the NBL farm group, compared to NBH and BED groups. However, lower space allowance per 100 kg of live weight (SK), was detected in NBH than in NBL and BED; differences are not, though by a small margin, statistically significant (*p* = 0.067) and show that the SK value decreases as the pig live weight increases, particularly in heavy pigs.

A higher number of pigs per drinker (PD) and lower percentage of pens with at least 1 nipple/12 pigs or 1 water bowl/15 pigs (PDC) were found (*p* = 0.003 and *p* = 0.044, respectively) in the NBH group, compared to the NBL group. Farms belonging to the BED group were characterized by higher prevalence (*p* = 0.001) of optimal or suboptimal enrichment (EC) and higher percentage of observed active pigs manipulating (EMB) and in reach (PAE) of enrichment materials (*p* = 6 × 10^−7^ and *p* = 0.001, respectively), compared to NBH and NBL farms.

The average number of pigs per farm (AVP) and per Annual Work Unit (PWU) in 2018, were found much lower (*p* = 9 × 10^−5^ and *p* = 0.024, respectively) in the BED farm group, than in the NBH and NBL farm groups. Regarding the productive performances, the average daily growth (ADG) was lower (*p* = 4 × 10^−4^) in NBH farms than in the NBL and BED farm groups, whereas the feed conversion rate (FCR) was statistically different in the three groups (*p* = 4 × 10^−7^) and higher NBH farms than in NBL and BED farms.

Average pig mortality rate and medication cost per pig sold were found statistically different from each other in the three farm groups (*p* = 0.003 and *p* = 0.046, respectively). No statistically significant difference between the farm percentages of observed pens with at least one tail lesion (T) in the three farm clusters (*p* = 0.363), although all pigs in the NBH and NBL groups were tail docked and most pigs in the BED group (i.e., except in one farm) were tail undocked. Ear and body lesions were detected more frequently in the NBL group than in the NBH and BED groups (*p* = 0.023 and *p* = 2 × 10^−4^, respectively).

Pigs with tails shortened by less than 50% of the total tail length (SHT) were found more common in NBL farms than in NBH farms, where most pig tails were shortened by more than 50% of the total length (STT) (Table 4). Dirtiness in the laying area was statistically different (*p* = 0.008) and higher in farms of the NBL group, compared to those if the BED group, but no statistical difference was found between the NBH and BED farm groups. Liquid feeding system was observed as less common in the BED group, compared to the other two farm groups (*p* = 0.001).

The organic farms in the dataset are all included in the BED group. Most farms allowing pigs to access outdoors are also in the BED farm group; only one of them belongs to the NBL group. Feed restriction is used in 90% of the NBH farms and in only 29% and 30% of the NBL and BED farms, respectively (Table 5).

## 4. Discussion

Lower farm size (AVP) and number of pigs per Annual Work Unit (PWU) were found in farms of the BED group, compared to farms of the other two clusters, suggesting that the higher labour needed per pig in these farms could be related to a higher workload for the management of bedding materials, as well as to less economies of scale in place in these smaller sized farms. These farms are featured by no or limited slatted floor, presence of proper manipulable material and roughage, higher space allowance and no or limited tail docking; almost all pigs observed in these farms were found in reach of optimal enrichment materials, according to the EC Recommendation 336/2016 [23], and most active pigs were manipulating them in the observed pens. Lower prevalence of ear and body lesions was found in these farms, compared to lean pig farms without bedded solid floored resting area (NBL), but not compared to heavy pig farms without bedded solid floored resting area (NBH), where the higher age of the observed pigs and the higher pig space allowance could have mitigated the occurrence of these lesions.

The group of heavy pig farms without bedded solid floored resting area (NBH) is characterized by the presence of fully or partially slatted floor, liquid feeding, and limited availability of drinkers, which is considered as a risk factor for pig welfare, particularly in summertime when the water nutritional need tends to increase [16], as well as the competition of pigs to access water.

Higher space allowance detected in the NBH and BED groups, compared to the HBL group, can be related to the higher live weight in NBH farms and the higher space allowance needed to house pigs with bedded solid floored resting area. It is worth noting that the pig space allowance of 1.15 m^2^/pig in the heavy pig farms of the NBH group exceeds the minimum legal requirement of 1 m^2^/pig for pigs over the live weight of 110 kg, although it is lower than the value on 1.48 m^2^/pig, calculated through the allometric formula for pigs of 172 kg of live weight with k = 0.047, as recommended by EFSA [10]. However, more space allowance for heavy pigs could be further used in the next few years by heavy pig farmers in an attempt to house pigs with intact tails, in compliance with EU legislation.

The lower presence of roughage and of bedding material in the laying area in farms of the NBH e and NBL groups, compared to the BED group, can also be related to the higher prevalence of slatted floor, which is likely to limit the use of a large quantity of organic materials (e.g., straw, wood shavings) because of the inability of the most common slurry systems (e.g., vacuum system) in place in European fattening pig farms to handle and evacuate these organic materials, together with the liquid manure under the slats, as confirmed in previous studies [2]. The relatively low dirtiness score in the laying area of NBH heavy pig farms can be related to the presence slatted floor in these farms.

Lean pig farms without bedded solid floored resting area (NBL) were featured by lower pig space allowance, mortality, medication cost, number of pigs per drinker and feed conversion rate and by higher average daily growth and prevalence of ear and body lesions, compared to NBH heavy pig farms.

No statistical difference was found between the farm percentages of observed pens with at least one tail lesion in one pig in the three farm groups, although tail docking was performed in both NBH and NBL farms on almost all pigs and not performed on most pigs in BED farms; this outcome suggests that the prevalence of tail biting in undocked tail pigs can be similar to docked tail pigs housed in intensive systems, if undocked tail pigs are housed with bedded solid floored resting area, plenty of manipulable material that most pigs are able to access, and larger space allowance above the minimum EU legal requirements.

Low prevalence of tail lesions in the NBH group can be related to the majority of pigs with tails shortened by more than 50% (STT), which is likely to expose the pigs to less severe tail lesions but also to more painful tail docking and more frequent formation of neuromas afterwards [26]. Low prevalence of tail lesions in NBH farms can be explained by the higher liveweight (AW) and age of the observed pigs, confirming the outcomes of a previous study showing more severe tail lesions in younger pigs, compared to older ones [39].

The observation of more prevalent ear and skin lesions in pigs of the NBL farm group could be ascribed to the lower space allowance at the start of the growing phase when more frequent fights may occur in recently grouped pigs to establish a hierarchy [40]. Higher pig mortality rates and medication costs per pig in NBH heavy pig farms can be explained by the longer duration of the fattening period and the higher age at slaughter of at least nine months in heavy pigs for Parma Ham PDO. Mortality rate and medication cost per pig could also be biased by different culling rates across farms, due to a different degree of implementing this practice to reduce animal suffering and the spread of diseases on farms.

## 5. Conclusions

In conclusion, the set of animal and non-animal-based measures used in this study, was found suitable and useful to assess, analyse, and compare most of the housing risk factors for pig welfare on farms. Greater validity of the statistical model used in this study could result from a greater availability in the future of pig farm cases in the SusPigSys database.

Additional non-animal-based measures of pig welfare could also be considered to assess the microclimate pig comfort and the presence on noxious gases and dust as risk factors for tail biting and for overall welfare assessment. Animal-based indicators were used to monitor pig welfare directly and to investigate the effect that the resources and management have on the animals. Both animal and non-animal-based measures provided different types of information, which are needed for routine official controls and are suitable for use in farm assurance schemes.

Housing risk factors for pig welfare, such as the lack of space allowance, bedding, and environmental enrichment, as well as the presence of fully slatted floor and the availability of drinkers to ensure pigs have permanent access to drinkable water are likely to become more challenging for pig farmers to keep pigs intensively with long undocked tails, once the ban of routine tail docking is finally applied across all EU Member States.

## Figures and Tables

**Figure 1 animals-11-03221-f001:**
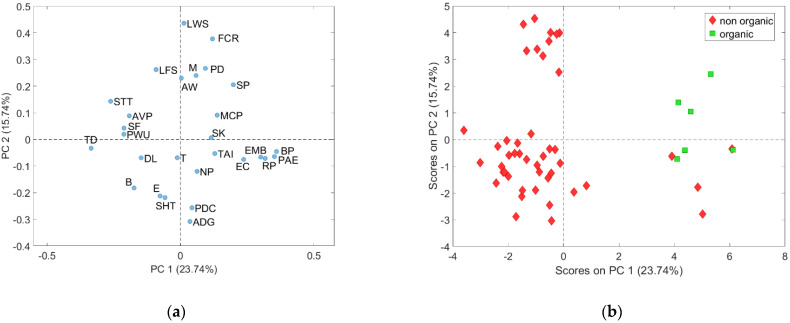
(**a**) Loading plot of PC1 and PC2; (**b**) Score plot of PC1 and PC2, representing organic and non-organic farms.

**Figure 2 animals-11-03221-f002:**
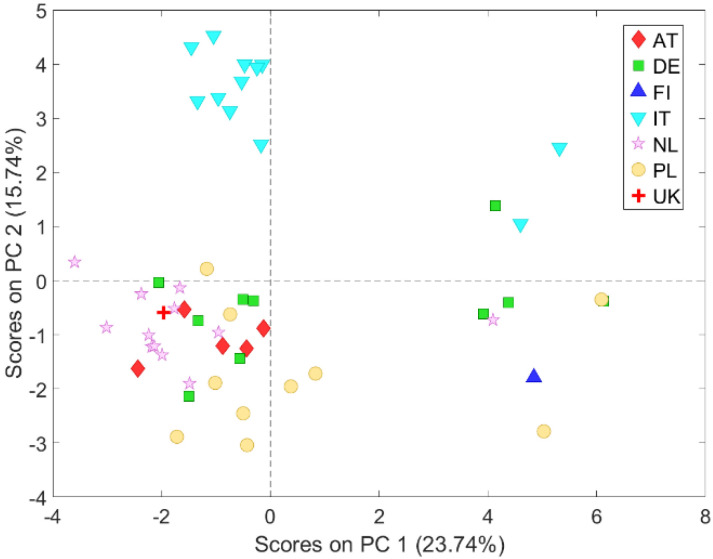
Score plot of PC1 and PC2; farm countries in evidence.

**Table 1 animals-11-03221-t001:** Continuous, ordinal, and dichotomous variables for 709 observed pig groups.

N.	Variable Description	Acronym	Type 1 ^1^	Type 2 ^2^
1	Total area indoor of observed pens	TAI	nABM	C
2	Number of pigs per observed pen	NP	nABM	C
3	Average pig live weight in observed pens	AW	ABM	C
4	Mean space allowance (m^2^) per pig in observed pens	SP	nABM	C
5	Mean space allowance per 100 kg of pig liveweight in observed pens	SK	ABM	C
6	Mean number of pigs per drinker in observed pens	PD	nABM	C
7	% of active pigs manipulating enrichment in observed pens	EMB	ABM	C
8	% of active pigs in reach of enrichment material in observed pens	PAE	ABM	C
9	Laying area dirtiness: 1 = clean; 2 = medium; 3 = dirty	DL	nABM	O
10	Slatted Floor: 0 = no; 1 = partial; 2 = totally slatted	SF	nABM	O
11	Liquid feeding system: 1 = dry; 2 = wet or mixed dry/liquid; 3 = liquid	LFS	nABM	O
12	Bedding in lying area: 0 = no bedding, 1 = not all pigs can lie on bedded area, 2 = enough in laying area; 3 = all pen floor bedded	BP	nABM	O
13	Presence of roughage: 0 = no roughage; 1 = pellet; 2 = straw; 3 = hay/silage	RP	nABM	O
14	Presence of enrichment: 0 = no enrichment; 1 = other enrichment of marginal interest; 2 = suboptimal or optimal/proper enrichment combined with other enrichment; 3 = proper/optimal enrichment	EP	nABM	O
15	Presence of tail docked pigs: 0 = no tail docked; 1 = some tail docked; 2 = all tail docked	TD	nABM	O
16	Short tail: 0 = no pigs with tail shortened by less than 50% of the original length; 1 = ≤10% pigs with tails shortened by less than 50%; 2 = >10% pigs with tails shortened by less than 50%	SHT	ABM	O
17	Tail stump: 0 = no pigs with tail shortened by more than 50% of the original length; 1 = ≤10% pigs with tails shortened by more than 50%; 2 = >10% pigs with tails shortened by more than 50%	STT	ABM	O
18	Tail lesions: 0 = no; 1 = ≤10% pigs have mild damage, but no pig has severe damage; 2 = >10% pigs have mild damage, and/or 1 has severe damage	T	ABM	O
19	Ear lesions: 0 = no; 1 = ≤10% pigs have mild damage, but no pig has severe damage; 2 = >10% pigs have mild damage, and/or 1 has severe damage	E	ABM	O
20	Body lesions: 0 = no skin lesions; 1 = ≤20% pigs have mild skin lesions, but no pig has severe damage; 2 = >20% pigs have mild skin lesions, and/or 1 has severe damage	B	ABM	O
21	Feed restriction 0 = no, 1 = yes	FR	nABM	D
22	Outdoor access 0 = no, 1 = yes	OA	nABM	D
23	Organic farm 0 = no, 1 = yes	OR	nABM	D

^1^ Animal Based Measure (ABM) or non-Animal Based Measure (nABM). ^2^ continuous (C) or ordinal (O) or dichotomous (D) variables.

**Table 2 animals-11-03221-t002:** Continuous, ordinal and dichotomous variables for 51 fattening pig units.

N.	Variable Description	Acronym	Type 1 ^1^	Type 2 ^2^
1	Farm mean total area indoor of observed pens	TAI	nABM	C
2	Farm mean number of pigs in observed pens	NP	nABM	C
3	Average pig live weight in observed pens	AW	ABM	C
4	Farm mean space allowance (m^2^) per pig in observed pens	SP	nABM	C
5	Farm mean space allowance per 100 kg of pig live weight in observed pens	SK	ABM	C
6	Farm mean number of pigs per drinker in observed pens	PD	nABM	C
7	Farm % of pens with at least 1 nipple per 12 pigs or 1 water bowl per 15 pigs	PDC	nABM	C
8	Farm mean % of active pigs manipulating enrichment in observed pens	EMB	ABM	C
9	Farm mean % of active pigs in reach of enrichment material in observed pens	PAE	ABM	C
10	Farm % of observed pens with optimal or suboptimal enrichment	EC	nABM	C
11	Farm average number of pigs per farm in 2018	AVP	nABM	C
12	Farm average number of pigs per Annual Work Unit in 2018	AWU	nABM	C
13	Farm maximum pig live weight before slaughter in 2018	LWS	nABM	C
14	Farm Average Daily Growth in 2018	ADG	ABM	C
15	Farm average Feed Conversion Rate in 2018	FCR	ABM	C
16	Farm veterinary and medication per pig sold in 2018 (EUR/pig)	MCP	nABM	C
17	Average mortality in 2018 (not including culled pigs)	M	ABM	C
18	Farm % of observed pens with at least one tail lesion in one pig	T	ABM	C
19	Farm % of observed pens with at least one ear lesion in one pig	E	ABM	C
20	Farm % of observed pens with at least one skin lesion in the body of one pig	B	ABM	C
21	Farm presence of tail docked pigs: 0 = no tail docked; 1 = some tail docked; 2 = all tail docked	TD	nABM	O
22	Farm presence of pigs with short tail: 0 = no pigs with short tail; 1 = ≤10% pigs with short tail; 2 = >10% pigs with short tail	SHT	ABM	O
23	Farm presence of pigs with tail stump: 0 = no pigs with tail stump; 1 = ≤10% pig with tail stump; 2 = >10% pigs with tail stump	STT	ABM	O
24	Farm presence of slatted floor: 0 = no slatted floor; 1 = partially slatted floor; 2 = total slatted floor	SF	nABM	O
25	Farm presence of bedding in lying area: 0 = no bedding, 1 = enough bedding in laying area; 2 = all pen floor bedded	BP	nABM	O
26	Farm presence of roughage: 0 = no roughage; 1 = pellet; 2 = straw; 3 = hay/silage	RP	nABM	O
27	Farm laying area dirtiness score: 1 = clean; 2 = medium; 3 = dirty	DL	nABM	O
28	Farm presence of liquid feeding system: 1 = dry; 2 = wet or mixed dry/liquid; 3 = liquid	LFS	nABM	O
29	Organic farm: 0 = no, 1 = yes	OR	nABM	D
30	Outdoor access: 0 = no, 1 = yes	OA	nABM	D
31	Feed restriction: 0 = no, 1 = yes	FR	nABM	D

^1^ Animal Based Measure (ABM) or non-Animal Based Measure (nABM) ^2^ continuous (C) or ordinal (O) or dichotomous (D) variables.

**Table 3 animals-11-03221-t003:** Statistics for 20 continuous variables.

Variables		NBH (10 Farms)			NBL (31 Farms)			BED (10 Farms)			
N.	Acronym	Q25	Mdn	Q75	Q25	Mdn	Q75	Q25	Mdn	Q75	*p*-Value
1	TAI	14.2	20.6	28.9	11.5	18.2	35.0	15.6	21.1	50.1	>0.05
2	NP	12.2	17.3	25.5	14.2	21.1	37.0	14.3	19.3	40.0	>0.05
3	AW	93.2	110.2 ^a^	117.4	62.0	70.0 ^b^	78.2	57.4	75.0 ^b^	90.8	<0.01
4	SP	1.11	1.15 ^a^	1.25	0.79	0.88 ^b^	1.01	1.03	1.20 ^a^	1.30	<0.01
5	SK	1.09	1.24	1.57	1.15	1.40	1.63	1.26	1.77	2.25	>0.05
6	PD	10.7	14.5 ^a^	25.5	7.2	8.5 ^b^	11.9	7.3	9.4 ^a,b^	17.6	<0.01
7	PDC	0.0	20 ^a^	80.6	33.3	93 ^b^	100.0	25.7	97 ^a,b^	100.0	<0.05
8	EMB	5.5	6.8 ^a^	13.6	4.8	12.1 ^a^	21.4	41.3	72.3 ^b^	87.4	<0.01
9	PAE	5.9	9.2 ^a^	11.8	10.5	16.1 ^b^	24.5	98.7	100 ^c^	100.0	<0.01
10	EC	0.0	33 ^a^	88.3	0.0	53 ^a^	100.0	100.0	100 ^b^	100.0	<0.01
11	AVP	1571	2169 ^a^	3226	976	1450 ^a^	2419	191	263 ^b^	100.0	<0.01
12	AWU	967	1187 ^a^	1355	444	1065 ^a^	2182	152	460 ^b^	915	<0.05
13	LWS	165	172 ^a^	175	118	120 ^b^	122	115	122 ^b^	140	<0.01
14	ADG	680	708 ^a^	753	800	820 ^b^	885	741	780 ^b^	1000	<0.01
15	FCR	3.6	3.7 ^a^	3.8	2.5	2.6 ^b^	2.8	2.8	3.0 ^c^	3.5	<0.01
16	MCP	1.4	2.6^a^	3.4	0.5	1.0 ^b^	2.8	1.1	2.0 ^b^	4.9	<0.05
17	M	3.3	3.8 ^a^	4.6	1.5	2.0 ^b^	2.9	2.0	2.7 ^b^	4.0	<0.01
18	T	0.0	3.3	6.7	0.0	6.7	20.0	0.0	6.7	21.7	>0.05
19	E	0.0	0.0 ^a^	1.7	0.0	6.7 ^b^	40.0	0.0	0.0 ^a^	0.7	<0.05
20	B	0.0	0.0 ^a^	3.3	6.7	38.5 ^b^	73.3	0.0	0.0 ^a^	8.3	<0.01

TAI, Total area indoor of observed pens; NP, pigs/pen; AW, average pig live weight/pen; SP, space allowance/pig; SK, space allowance/100 kg pig LW; PD, pigs/drinker; PDC, % of pens with at least 1 nipple/12 pigs or 1 water bowl/15 pigs; EMB, % of active pigs manipulating enrichment; EMB, % of active pigs manipulating enrichment; PAE, % of active pigs in reach of enrichment; EC, % of pens with optimal or suboptimal enrichment; AVP, average number of pigs/farm; AWU, average number of pigs/AWU; LWS, pig live weight at slaughter; ADG, average daily growth; FCR, feed conversion rate; MCP, medication cost/pig; M, mortality rate; T, % of pens with at least one pig tail lesion; E, % of pens with at least one ear lesion; B, % of pens with at least one skin lesion. Median (Mdn), lower quartile (Q25) and upper quartile (Q75) values for assessed measures per housing system (i.e., BED, NBL and NBH). *p* = result of global Kruskal–Wallis test for housing system effect. ^a, b, c^ Median values with different superscripts within a row differ at *p* < 0.05 in a pairwise system comparison with Mann–Whitney U-test.

**Table 4 animals-11-03221-t004:** Statistics for 8 ordinal variables.

Variables		NBH (10 Farms)			NBL (31 Farms)			BED (10 Farms)			
N.	Acronym	Q25	Mdn	Q75	Q25	Mdn	Q75	Q25	Mdn	Q75	*p*-Value
1	TD	1.8	2.0 ^a^	2.0	2.0	2.0 ^a^	2.0	0.0	0.0 ^b^	0.0	<0.01
2	SHT	0.0	0.0 ^a^	1.0	0.0	2.0 ^b^	2.0	0.0	0.0 ^a^	0.0	<0.01
3	STT	2.0	2.0 ^a^	2.0	0.0	2.0 ^a^	2.0	0.0	0.0 ^b^	0.0	<0.01
4	SF	1.0	1.5 ^a^	2.0	1.0	1.0 ^a^	2.0	0.0	0.5 ^b^	1.0	<0.01
5	BP	0.0	0.0 ^a^	0.0	0.0	0.0 ^a^	0.0	2.0	3.0 ^b^	3.0	<0.01
6	RP	0.0	0.0 ^a^	0.0	0.0	0.0 ^a^	0.0	2.0	2.0 ^b^	2.3	<0.01
7	DL	0.0	0.0 ^a,b^	0.3	0.0	1.0 ^a^	1.0	0.0	0.0 ^b^	0.0	<0.01
8	LFS	2.0	2.0 ^a^	2.0	0.0	0.0 ^a^	2.0	0.0	0.0 ^b^	1.3	<0.01

TD, tail docked pigs; SHT, pigs with short tails; STT, pigs with tail stump; SF, slatted floor; BP, presence of bedding in the laying area; RP, presence of roughage; DL, dirtiness score in the laying area; LFS, liquid feeding system. Median (Mdn), lower quartile (Q25) and upper quartile (Q75) values for assessed measures per housing system (i.e., BED, NBL and NBH). *p* = result of global Kruskal–Wallis test for housing system effect. ^a, b^ Median values with different superscripts within a row differ at *p* < 0.05 in a pairwise system comparison with Mann–Whitney U-test.

**Table 5 animals-11-03221-t005:** Descriptive statistics for 3 dichotomous variables; percentage of farms in the group.

N.	Variable Description	Acronym	NBH %	NBL %	BED %
1	Organic farm	OR	0	0	60
2	Outdoor access	OA	0	3	70
3	Feed restriction	FR	90	29	30

## Data Availability

The data presented in this study are openly available in FigShare at https://doi.org/10.6084/m9.figshare.16726567 (accessed on 9 October 2021).

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
