# Peer review of "Analysis of Housing Risk Factors for the Welfare of Lean and Heavy Pigs in a Sample of European Fattening Farms"

_animals, 2021, doi:10.3390/ani11113221_

Round 1
Reviewer 1 Report
This paper analyses data from a very large study of pig performance and pig welfare in farms throughout Europe using a range of animal and non-animal measures as recommended by the Welfare Quality protocol. It provides a good picture of the range of conditions existing within the European pig industry.
Principle Component Analysis (PCA) has been used to identify correlations between the many variables and, from these, to identify three distinct management systems: NBH, NBL and BED. Having established the significant differences between these groups, the paper then examines the consequences of these differences for pig welfare (Table 3).
The whole of the results section is given to the exposition of the PCA (l344-437 and Figs 1-4) and includes many statements of the obvious (e.g. NBH pigs were heavier than NBL, TAI correlated with NP, AW with FCR). None of this is relevant to animal welfare. The important results are given in Table 3, which is wrongly placed in the discussion section. Table 3 needs to be viewed alongside Table 2, which lists the variables and the metrics used for each.
I recommend that the description of the PCA should be radically reduced to no more than that necessary to justify the allocation of the units to three groups. Table 3 should be included in the results section and discussion restricted to issues of relevance to animal welfare. Obvious differences (e.g. AW, SP, PAE) are superfluous and dilute the things that matter.
The introduction is much too long. It reads like a PhD thesis. It should radically abridged and restricted simply to that necessary to justify nd explain the purpose of this specific study.
There are a number of relatively minor points in the text that raise concern. I can deal with these specifics when I read the heavily abridged revision.
Author Response
Point 1: The whole of the results section is given to the exposition of the PCA (l344-437 and Figs 1-4) and includes many statements of the obvious (e.g. NBH pigs were heavier than NBL, TAI correlated with NP, AW with FCR). None of this is relevant to animal welfare. The important results are given in Table 3, which is wrongly placed in the discussion section. Table 3 needs to be viewed alongside Table 2, which lists the variables and the metrics used for each.
Response 1: I omitted the statements of the obvious. I divided Table 3 in two Tables 3 and 4 and placed them the Results section, provided with footnote with the meaning of the acronyms to facilitate the reading.
Point 2: I recommend that the description of the PCA should be radically reduced to no more than that necessary to justify the allocation of the units to three groups. Table 3 should be included in the results section and discussion restricted to issues of relevance to animal welfare. Obvious differences (e.g. AW, SP, PAE) are superfluous and dilute the things that matter.
Response 2: Description of PCA is radically reduced. Comments about obvious differences omitted.
Point 3: The introduction is much too long. It reads like a PhD thesis. It should radically abridged and restricted simply to that necessary to justify nd explain the purpose of this specific study.
Response 3: You are right. The introduction is from my PhD Thesis. Sorry. I did my best to abridge and restrict it to that necessary to justify and explain the purpose of this study. The long introduction was motivated by the high number of variables in this study.
Point 4: There are a number of relatively minor points in the text that raise concern. I can deal with these specifics when I read the heavily abridged revision.
Response 4: The revised manuscript is ready now for your further revision
Reviewer 2 Report
ANIMALS 1435383
Overall this is an interesting study which appears to be well-designed and provides new information on animal welfare and its assessment.
I find the introduction extremely detailed and long. Perhaps some information presented in the Introduction could be moved to the Discussion?
Lines 316-333: Please check, there is a formatting issue in those lines.
Line 441: Please check, there is a misspelling liveweight must say live weight
Line 441: Consider including acronyms for live weight at slaughter LWS
Line 457: There is a Typo “he higher average live weight in NBH” should say: The higher average
Lines 469-492: Please check, there is a formatting issue in those lines.
Lines 506-507: could you please check this sentence? It would seems that there is something missing “…although tail docking was performed in all NBH and NBL farms and in only farm of the BED group.”
Line 547: Please remove the extra blank space “suggesting that the”
Line 550: Please remove the extra blank space “space allowance”
Line 550: Please check, there is a misspelling liveweight must say live weight
Lines 566-569: Please check this paragraph “The herd size and the number of pigs per Annual Work Unit was much lower in the BED group than in the other two farm groups, suggesting that the higher labor need in these farms could be related to an higher work load for the management of bedding materials and to less economies of scale in place in smaller farms.”
Table 1
There is a misspelling in N 15 (page 7) the word “length”.
I would suggest to align the information under variable description to the left, it would be easier to read
Table 2
Please check N 13 there is a misspelling liveweight must say live weight
Please check N 26 the word presence is repeated: Farm presence of roughage presence
I would suggest to align the information under variable description to the left, it would be easier to read
Figures 1, 2 and 3 look blurry, There is any way to make them clearer?
Table 3
If you will need to divide the table in two pages, I suggest to repeat the headline this will make the table easier to read.
Table 4
I would recommend to include a footnote with the meaning of the acronyms, this would facilitate the reading of the table.
Suggestion: consider changing P values, standardize to P<0.05, P>0.05, or P<0.01
Conclusions: Part of conclusion corresponds to the discussion, please consider moving lines 525 to 569 to section 4 Discussions.
Author Response
Point 1: I find the introduction extremely detailed and long. Perhaps some information presented in the Introduction could be moved to the Discussion?
Response 1: Done
Point 2: Lines 316-333: Please check, there is a formatting issue in those lines.
Response 2: Sorry, I checked it but I didn’t find that issue. However, I formatted Lines 316-333 again.
Point 3: Line 441: Please check, there is a misspelling liveweight must say live weight
Response 3: Checked and corrected
Point 4: Line 441: Consider including acronyms for live weight at slaughter LWS
Response 4: Done
Point 5: Line 457: There is a Typo “he higher average live weight in NBH” should say: The higher average
Response 5: Done
Point 6: Lines 469-492: Please check, there is a formatting issue in those lines.
Response 6: Done
Point 7: Lines 506-507: could you please check this sentence? It would seems that there is something missing “…although tail docking was performed in all NBH and NBL farms and in only farm of the BED group.”
Response 7: I rephrased the sentence
Point 8: Line 547: Please remove the extra blank space “suggesting that the”
Response 8: Done
Point 9: Line 550: Please remove the extra blank space “space allowance”
Response 9: Done
Point 10: Line 550: Please check, there is a misspelling liveweight must say live weight
Response 10: Done
Point 11: Lines 566-569: Please check this paragraph “The herd size and the number of pigs per Annual Work Unit was much lower in the BED group than in the other two farm groups, suggesting that the higher labor need in these farms could be related to an higher work load for the management of bedding materials and to less economies of scale in place in smaller farms.”
Response 11: I rephrased the sentence and replaced “higher labor need” with “higher labor need per pig”
Point 12: Table 1. There is a misspelling in N 15 (page 7) the word “length”. I would suggest to align the information under variable description to the left, it would be easier to read
Response 12: I replaced the wrong word “length” with the correct word “length”. You are right but the text in the table is formatted according to the template from Instructions for authors. I would be happy to do this change, if possible.
Point 13: Table 2. Please check N 13 there is a misspelling liveweight must say live weight
Please check N 26 the word presence is repeated: Farm presence of roughage presence. I would suggest to align the information under variable description to the left, it would be easier to read. Figures 1, 2 and 3 look blurry, There is any way to make them clearer?
Response 13: I replaced liveweight with live weight. I cancelled the repetition of presence. You are right but the text in the table is formatted according to the template from Instructions for authors. I would be happy to do this change, if possible. I reworked the figures to make them clearer
Point 14: Table 3. If you will need to divide the table in two pages, I suggest to repeat the headline this will make the table easier to read.
Response 14: Done
Point 15: Table 4. I would recommend to include a footnote with the meaning of the acronyms, this would facilitate the reading of the table. Suggestion: consider changing P values, standardize to P<0.05, P>0.05, or P<0.01
Response 14: Done
Point 15: Conclusions: Part of conclusion corresponds to the discussion, please consider moving lines 525 to 569 to section 4 Discussions.
Response 15: Done
Reviewer 3 Report
General feedback
This is an interesting paper using a recently validated protocol (SusPigSys) for the on-farm evaluation of pig welfare based on resource-, management- and animal-based indicators. The study presents a comparison of EU farms housing pigs under very different conditions in relation to the risk factors they present for animal welfare. Some of the insights gained with this study might be useful for improving EU standards for pig welfare on farm. In this respect, I think that the paper is worthy of publication after a moderate revision. You will find below my detailed comments and suggestions.
Note
Symbols used
">>" means a suggested change
Detailed feedback
L 45 please provide reference to the legislation mentioned here (1991)
L 47 I think that the reference to "bulky feed" needs to be further clarified for the readers: could you please specify the problem associated with (the lack of) bulky feed? Are you referring here to the role of bulky feed as enrichment?
L 49-50 I suggest to rephrase for added clarity "The routine tail docking of pigs is banned under EU law (REF to the law) but allowed..."
L 52-53 I am afraid I have to challenge this statement. The solutions clearly exist as many farmers are rearing pigs with intact tails, notably in Finland, where pig farming is typically intensive, but also in many other EU countries, including a handful of producers in countries in Southern Europe, where conditions are even less favourable to providing enrichment. There is also a wealth of resources made available to disseminate best practice (see the resources on the website of the European Commission). The problem, as the title of your reference states, is systemic, meaning that there is no real incentive for farmers to change practices - no consistency in applying dissuasive measures at national and EU level (no infringements proceedings so far) and no penalisation at slaughter for farmers who consistently dock tails. I would suggest to rephrase this statement to better reflect the current EU reality.
L 55-56 Please nuance this statement. "Customers" is generic and might be interpreted as meaning "consumers", whereas the costs of improving animal welfare (and as a matter of fact here we are just talking about implementing EU basic legislation...) should be equitably distributed along the whole supply chain.
L 63 It is not a scheme but a protocol, to the best of my knowledge
L 75 >> "in the on-farm evaluation of..."
L 81 >> "not only one"
L 86 advisement = advisory?
L 74-89 I think it would be useful to make a distinction here. Typically, lawa (national and EU) have prescriptions that are only based on resources and management (the situation is likely to change as EU legislators are slowly accepting the idea that animal-based measures are important indicators of animal welfare). Beyond resource and management-based measures, most farm assurance schemes incorporate several animal-based indicators because ultimately what is important is the effect that the resources + management have on the animals. For instance, the EFSA clearly stated that an intact tail is perhaps the most important animal welfare indicator for pigs. Welfare Quality protocol has many other animal-based indicators. As you rightly say, it is important to have both as they provide different types of information. It is necessary to check resource and management-based indicators as part of routine official controls. Additionally, it is meaningful to also check animal-based indicators to have a sense of whether the resources provided result in good lives for the animals.
L 90 I would create a sub-section here (same as Introduction) for added clarity for readers. You could use "Risk factors for decreased animal welfare on pig farms" or something similar - just a suggestion!
L 150 "Front skin lesions": please clarify (shoulders, ears and face?)
L 152-153 I would nuance this statement as there is only one reference to support it (or add more references) - something like "one study found...so special attention should be given to providing effective enrichment as the pigs' live weight increases".
L 160-161 Please see my comment below.
L 181-184 I would put this paragraph at L 164 (after ref 11) to explain that solutions are available to provide pigs on fully slatted floors with some straw as environmental enrichment.
L 189 As you rightly note, mortality rate is not a risk factor but an indicator (= the outcome of several unaddressed risk factors...) so possibly not appropriate here because you are listing the risk factors. Maybe put it for last?
L 253 animal welfare >> animal-based
L 277, 281, 284, 287, 290 be mindful of consistency in the use of verb tenses ("were" vs. "have been"). Simple past throughout is preferable, IMO.
L 285 "grouped" = correct word?
L 288 "places" = do you mean areas/pens in the farm? If so, please specify
L 293-294 Could you specify the measured inter- and intra-observer agreement after the training session here, please?
L 295-97 Please be mindful that it is considered increasingly important to obtain green light from the institutional Ethics Committee and include ethics statements when interviewing people. In the absent of an Ethics Committee authorisation number, I would advise to at least better specify your process with a dedicated sentence in the M&M or in a dedicated sub-section. See, for example:
- Authors must state that written informed consent was obtained from the participants of the study (and the relevant document(s) must be provided when requested by the journal). If verbal informed consent was obtained, the reason(s) for the absence of written consent must be provided.
Other potentially important elements to specify
- Informed consent;
- Respect the confidentiality and anonymity of your research respondents
- Ensure that your participants will participate in your study voluntarily
Table 1 & 2
Measure n. 5 why was this measure considered an ABM?
Measure n. 15 why is this measure nABM?
Table 2
Measure n.8 why was this measure considered a nABM?
Measure n. 10 why was this measure considered an ABM? Isn't this a resource-based measure?
Measure 21 why is this measure a nABM, whereas 22 and 23 are (rightly) ABMs?
L 317 six ones>> six ones
L 322 Is this questionnaire part of the SusPigSys protocol? Otherwise it needs to be included as supplementary material
L 324-26 "Stockman" >> stockperson (many stockpersons are female)
L 332 that >> ,which
L 345-352 I think this is still part of M&Ms and not a result
Figures
Please note that the Figures have a very low resolution in the PDF file and are therefore difficult to read. The Figures need to be reworked so that they have a higher resolution in the final version of the paper.
Discussion & Conclusions
Before giving more detailed feedback on this section, I would recommend that the authors made a clearer separation between the results and the discussion proper. This discussion still includes too many results, which should be included in the appropriate section.
On the other hand, many of the considerations included in the conclusions are best suited to the discussion section.
I would kindly ask the authors to rearrange the arguments so that they belong to the appropriate sections. In my opinion, this will improve the readers' experience.
The Conclusions should be shortened and this can be done by moving some of the arguments to the discussion section, which at the moment is still too focused on the results. Furthermore, I would be keen for the authors to further clarify their recommendations based on their preliminary findings, if they feel comfortable and confident doing so.
Author Response
Point 1: L 45 please provide reference to the legislation mentioned here (1991)
Response 1: Done
Point 2:L 47 I think that the reference to "bulky feed" needs to be further clarified for the readers: could you please specify the problem associated with (the lack of) bulky feed? Are you referring here to the role of bulky feed as enrichment?
Response 2: I skipped out bulky feed from this sentence as it is relevant for pregnant sows (i.e. much less for fattening pigs). I also omitted early weaning and pig castration as these practices were not investigated in this study.
Point 3: L 49-50 I suggest to rephrase for added clarity "The routine tail docking of pigs is banned under EU law (REF to the law) but allowed..."
Response 3: I have deleted this sentence in the attempt to reduce the text of the introduction as much as possible, as requested by the other two reviewers. Tail docking is dealt with below in general although I agree with you that the issue of the application of European legislation on tail docking needs more room to be explained, which is not available in this article, unfortunately.
Point 4: L 52-53 I am afraid I have to challenge this statement. The solutions clearly exist as many farmers are rearing pigs with intact tails, notably in Finland, where pig farming is typically intensive, but also in many other EU countries, including a handful of producers in countries in Southern Europe, where conditions are even less favourable to providing enrichment. There is also a wealth of resources made available to disseminate best practice (see the resources on the website of the European Commission). The problem, as the title of your reference states, is systemic, meaning that there is no real incentive for farmers to change practices - no consistency in applying dissuasive measures at national and EU level (no infringements proceedings so far) and no penalisation at slaughter for farmers who consistently dock tails. I would suggest to rephrase this statement to better reflect the current EU reality.
Response 4: I deleted this sentence in the attempt to reduce the text of the introduction as much as possible, as requested by the other two reviewers.
Point 5: L 55-56 Please nuance this statement. "Customers" is generic and might be interpreted as meaning "consumers", whereas the costs of improving animal welfare (and as a matter of fact here we are just talking about implementing EU basic legislation...) should be equitably distributed along the whole supply chain.
Response 4: same as response 4
Point 5: L 63 It is not a scheme but a protocol, to the best of my knowledge
Response 5: You are right. I corrected it and rephrased the sentence.
Point 6: L 75 >> "in the on-farm evaluation of..."
Response 6: same as response 4
Point 7: L 81 >> "not only one"
Response 7: same as response 4
Point 8: L 86 advisement = advisory?
Response 8: same as response 4
Point 9: L 74-89 I think it would be useful to make a distinction here. Typically, lawa (national and EU) have prescriptions that are only based on resources and management (the situation is likely to change as EU legislators are slowly accepting the idea that animal-based measures are important indicators of animal welfare). Beyond resource and management-based measures, most farm assurance schemes incorporate several animal-based indicators because ultimately what is important is the effect that the resources + management have on the animals. For instance, the EFSA clearly stated that an intact tail is perhaps the most important animal welfare indicator for pigs. Welfare Quality protocol has many other animal-based indicators. As you rightly say, it is important to have both as they provide different types of information. It is necessary to check resource and management-based indicators as part of routine official controls. Additionally, it is meaningful to also check animal-based indicators to have a sense of whether the resources provided result in good lives for the animals.
Response 9: I agree with you that more discussion could be included in this paper to illustrate current and future use of resource based and animal based measure in farm assurance schemes. Unfortunately, I’m asked to reduce the introduction as much as possible. However, I included a statement about this distinction in the conclusions.
Point 10: L 90 I would create a sub-section here (same as Introduction) for added clarity for readers. You could use "Risk factors for decreased animal welfare on pig farms" or something similar - just a suggestion!
Response 10: Your suggestion is good but I did not follow it because: I radically reduced the introduction; I don’t see possibility of subsection in the Introduction according to Animals Instructions for authors.
Point 11: L 150 "Front skin lesions": please clarify (shoulders, ears and face?)
Response 11: : same as response 4.
Point 12: L 152-153 I would nuance this statement as there is only one reference to support it (or add more references) - something like "one study found...so special attention should be given to providing effective enrichment as the pigs' live weight increases".
Response 12: Done
Point 13: L 160-161 Please see my comment below.
Response 13: I deleted this sentence in the attempt to reduce the text of the introduction as much as possible, as requested by the other two reviewers. However the concept was introduced already in relation to slatted floor.
Point 14: L 181-184 I would put this paragraph at L 164 (after ref 11) to explain that solutions are available to provide pigs on fully slatted floors with some straw as environmental enrichment.
Response 14: I kept this sentence in the same place because I deleted L160-161.
Point 15: L 189 As you rightly note, mortality rate is not a risk factor but an indicator (= the outcome of several unaddressed risk factors...) so possibly not appropriate here because you are listing the risk factors. Maybe put it for last?
Response 15: Done.
Point 16: L 253 animal welfare >> animal-based
Response 16: Done
Point 17: L 277, 281, 284, 287, 290 be mindful of consistency in the use of verb tenses ("were" vs. "have been"). Simple past throughout is preferable, IMO.
Response 17: Thank you for reminding. I corrected a number of inconsistencies
Point 18: L 285 "grouped" = correct word?
Response 18: Yes, but I made the sentence clearer (i.e. one third of the pig groups at the start of the fattening period but grouped at least two weeks before farm visit, one third in the middle of the fattening period and one third at the end of the fattening period.
Point 19: L 288 "places" = do you mean areas/pens in the farm? If so, please specify
Response 19: Places are places in the pen. This was an instruction not mentioned in the SusPigSys protocol. In practice pig observation was performed by stopping and walking inside the pens for looking at all pig to detect pig lesions 50 cm away from pigs. For this reason, I prefer to omit this “instruction”.
Point 20: L 293-294 Could you specify the measured inter- and intra-observer agreement after the training session here, please?
Response 20: I specified in the text that Inter Observer Reliability (IOR) was calculated as exact agreement between two observers and expressed as weighted Kappa, PABAK and percentage agreement.
Point 21: L 295-97 Please be mindful that it is considered increasingly important to obtain green light from the institutional Ethics Committee and include ethics statements when interviewing people. In the absent of an Ethics Committee authorisation number, I would advise to at least better specify your process with a dedicated sentence in the M&M or in a dedicated sub-section. See, for example:
Response 21: I added more information about this. To this regard, I already send to the Editor a signed statement in which I declared that farmers were provided with information sheets, including information about anonymity, why the research was being conducted, how their data were being used and if there were any risks associated, and were asked to return a signed informed consent before the start of data collection, in compliance with Regulation (EU) 2016/679 of the European Parliament and of the Council.
Point 22: Authors must state that written informed consent was obtained from the participants of the study (and the relevant document(s) must be provided when requested by the journal). If verbal informed consent was obtained, the reason(s) for the absence of written consent must be provided.
Other potentially important elements to specify
Informed consent;
Respect the confidentiality and anonymity of your research respondents
Ensure that your participants will participate in your study voluntarily
Response 22: Done
Point 23: Table 1 & 2. Measure n. 5 why was this measure considered an ABM? Measure n. 15 why is this measure nABM?
Response 23: I considered Measure n. 5 “Farm mean space allowance per 100 kg of pig live weight in observed pens” as ABM because it is calculated in relation to the ABM “Average pig live weight in observed pens” (i.e. SK=100/AW*SP). In Table 1 I considered “Presence of tail docked pigs” as a nABM because related to management. In Table 2 I considered Measure n. 15 “Feed Conversion Rate” as an ABM
Point 24: Table 2. Measure n.8 why was this measure considered a nABM? Measure n. 10 why was this measure considered an ABM? Isn't this a resource-based measure? Measure 21 why is this measure a nABM, whereas 22 and 23 are (rightly) ABMs?
Response 24: Sorry. Measure n. 8 is an ABM and Measure n. 10 is a nABM. I corrected them. Thank you. I considered Measure 21 “Farm presence of tail docked pigs” a nABM because it was taken by asking the farmer about the tail docking practice (i.e. management) whereas I considered Measures 22 and 23 as ABM because they were taken through pig observation.
Point 25: L 317 six ones>> six ones
Response 25: I replaced “six ones” with “six of them”
Point 26: L 322 Is this questionnaire part of the SusPigSys protocol? Otherwise it needs to be included as supplementary material
Response 26: Yes it is
Point 27: L 324-26 "Stockman" >> stockperson (many stockpersons are female)
Response 27: Done
Point 28: L 332 that >> ,which
Response 28: Done
Point 29: L 345-352 I think this is still part of M&Ms and not a result
Response 29: Done
Point 30: Figures. Please note that the Figures have a very low resolution in the PDF file and are therefore difficult to read. The Figures need to be reworked so that they have a higher resolution in the final version of the paper.
Response 30: Done
Point 31: Discussion & Conclusions
Before giving more detailed feedback on this section, I would recommend that the authors made a clearer separation between the results and the discussion proper. This discussion still includes too many results, which should be included in the appropriate section.
On the other hand, many of the considerations included in the conclusions are best suited to the discussion section.
I would kindly ask the authors to rearrange the arguments so that they belong to the appropriate sections. In my opinion, this will improve the readers' experience.
Response 31: Done
Point 32: The Conclusions should be shortened and this can be done by moving some of the arguments to the discussion section, which at the moment is still too focused on the results. Furthermore, I would be keen for the authors to further clarify their recommendations based on their preliminary findings, if they feel comfortable and confident doing so.
Response 32: Done
Round 2
Reviewer 1 Report
You have given proper attention to my criticism and suggestions, which mostly related to matters of presentation. The trial and analysis are sound and, in my opinion, now acceptable for publication.